# Physically Regularized Machine Learning Emulators of Aerosol Activation

Sam J Silva[1], Po-Lun Ma[1], Joseph C. Hardin[1], Daniel Rothenberg[2]

[1]Pacific Northwest National Laboratory, Richland, WA

[2]ClimaCell, Boston, MA

*Correspondence to*: Sam J. Silva (sam.silva@pnnl.gov)

**Abstract.** The activation of aerosol into cloud droplets is an important step in the formation of clouds, and strongly influences the radiative budget of the Earth. Explicitly simulating aerosol activation in Earth system models is challenging due to the computational complexity required to resolve the necessary chemical and physical processes and their interactions. As such,

various parameterizations have been developed to approximate these details at reduced computational cost and accuracy. Here, we explore how machine learning emulators can be used to bridge this gap in computational cost and parameterization accuracy. We evaluate a set of emulators of a detailed cloud parcel model using physically regularized machine learning regression techniques. We find that the emulators can reproduce the parcel model at higher accuracy than many existing parameterizations. Furthermore, physical regularization tends to improve emulator accuracy, most significantly when

emulating very low activation fractions. This work demonstrates the value of physical constraints in machine learning model development and enables the implementation of improved hybrid physical-machine learning models of aerosol activation into next generation Earth system models.

## 1 Introduction

Aerosols are important components of the Earth System, where they play a critical role in cloud processes and strongly

modulate the radiative budget. Aerosols impact radiation through directly absorbing and scattering light (Wallace and Hobbs, 2006), and by changing the radiative characteristics, lifetime, and abundance of clouds through a wide array of aerosol-cloud interactions (e.g. Albrecht, 1989; Twomey, 1977). This large influence is in-part because cloud formation through the direct condensation of atmospheric water vapor into cloud droplets is thermodynamically unfavorable in the atmosphere. Instead, cloud formation is largely initiated by the nucleation of cloud droplets through heterogenous interactions between water vapor

and aerosol (Seinfeld and Pandis, 2016). These aerosol-cloud interactions are the dominant radiative impact of aerosol in the Anthropocene, attributable to a large portion of the anthropogenic radiative forcing of aerosol (Bellouin et al., 2020). This influence on the global radiative budget is relatively large, and potentially offsets much of the warming associated with anthropogenic greenhouse gas emissions. Despite the importance of aerosol-cloud interactions, modern Earth System Models struggle to accurately represent these processes (Seinfeld et al., 2016). In total, the accurate simulation of aerosol-cloud

interactions is one of the largest uncertainties in modern Earth System Models, and a limiting factor in developing a predictive capability for the Earth System (Bellouin et al., 2020; Boucher et al., 2013; Committee on the Future of Atmospheric Chemistry Research et al., 2016).

Aerosol activation, also known as droplet nucleation, is a necessary step in the processes driving aerosol-cloud interactions. Once activated, aerosol can directly influence cloud properties (Wallace and Hobbs, 2006). For example, addition of activated aerosol to existing clouds can change the number concentration of cloud droplets, which impacts cloud brightness and lifetime, and the resulting net radiative impact of clouds (e.g. Christensen et al., 2020; Twomey, 1974, 1977). The aerosol activation process occurs at a scale much smaller than Earth System Model grid spacing and interacts with a variety of other sub-grid scale processes relating to clouds (e.g., turbulent mixing, convection, etc.).

Current methods for simulating aerosol activation require tradeoffs between model fidelity and computational efficiency. The most accurate models of aerosol activation simulate the thermodynamic and chemical conditions within a zero-dimensional parcel of air to analytically predict the fraction of aerosol activated into cloud droplets (e.g. Ghan et al., 2011; Rothenberg & Wang, 2015). These so-called "parcel models" explicitly resolve the condensational processes across a size resolved distribution of an aerosol population for a specified amount of time to calculate both the maximum supersaturation of the local atmosphere, and the total number of aerosols activated into cloud droplets. These parcel models are too computationally expensive to be used in global Earth System Models, and so various parameterizations have been developed with reduced computational cost.

Early parameterizations of aerosol activation were based on a few observations (e.g. Twomey, 1959) and were generally simple functions of a limited number of parameters. As computational and observational capabilities increased, these parameterizations increased in complexity (e.g. Abdul-Razzak & Ghan, 2000; Fountoukis & Nenes, 2005; Ming et al., 2006). Though these parameterizations all generally aim to calculate similar quantities, there are key differences in their implementation. These differences are largely based around the level of explicit versus approximated process-level details and degree of tuning within the parameterizations, described further for a variety of popularly used parameterizations in Ghan et al. (2011). The majority of these modern parameterizations perform similarly well for common atmospheric conditions, though relatively large differences (~30%) can be found in certain scenarios (Ghan et al., 2011). Despite their increased computational complexity, these parameterizations are still unable to fully reproduce the results of detailed parcel models, with errors on the order of ~10% (Rothenberg and Wang, 2015). While small, these errors can potentially compound in models with longer run times, further motivating development of emulators with improved predictive skill.

Recent work applying machine learning techniques to the emulation of computationally expensive systems has shown promise toward developing emulators that are both fast and accurate (e.g. Brenowitz & Bretherton, 2018; Gentine et al., 2018; Rasp et

al., 2018; Silva et al., 2020a). This is particularly the case for the class of so-called "physically informed" machine learning emulators (e.g. Raissi et al., 2019; Reichstein et al., 2019). Physically informed machine learning algorithms directly incorporate physical information into their construction and/or training with the aim of creating emulators that are more performant in terms of ease of training or resulting accuracy. For example, physical information can be encoded through penalizing emulators that violate known constraints (e.g. conservation of energy) in the cost function that is optimized during

model parameter optimization (e.g. Beucler et al., 2019). More complex methods of including physical information can be achieved through directly altering the architecture of a machine learning model to analytically enforce various constraints or follow a physically based model system (e.g. Zhao et al., 2019). These approaches of including physical information in machine learning model development ultimately have what is known as a "regularizing effect" on the model, wherein they help reduce model overfitting.

In this study, we explore how a hybrid physical and machine learning modeling approach can be used to develop improved parameterizations of aerosol activation. We demonstrate that applying a very simple constraint to commonly used machine learning techniques improves their predictive skill and can lead to more accurate and trustworthy emulators of computationally expensive models. We build on previous work (e.g. Rothenberg and Wang, 2015; Lipponen et al., 2013) by considering a wider array of emulator design methods and physical constraints.

**2 Modeling Approach**

We develop emulators of a detailed parcel model for aerosol activation. Specifically, we train several classes of machine learning models to emulate the "Pyrcel" parcel model, reproducing the fraction of aerosol particles activated (Rothenberg & Wang, 2015). Pyrcel simulates the activated fraction of an initial population of aerosol in a zero-dimensional parcel of air as it adiabatically raises in the atmosphere. Here, we use a single aerosol mode with 250 bins and an initial supersaturation of

zero. This broadly assumes that we are making these calculations directly at the base or edge of a cloud. Other atmospheric conditions used as inputs to the Pyrcel model are varied to generate the emulator development datasets and are summarized in Table 1. For more details on the Pyrcel model, see Rothenberg & Wang (2015).

| Quantity | Units | Range | Name |
|---|---|---|---|
| $Log_{10}N$ | $Log_{10}cm^{-3}$ | [1, 4] | Mode Number Concentration |
| $Log_{10}ug$ | $Log_{10}um$ | [-3, 1] | Mode Geometric Mean Radius |
| $Sigma_g$ | - | 1.6 or 1.8 | Mode Standard Deviation |
| Kappa | - | [0, 1.2] | Mode Hygroscopicity |
| $Log_{10}V$ | $Log_{10}ms^{-1}$ | [-2, 1] | Updraft Velocity |
| T | K | [248, 310] | Air Temperature |
| P | Pa | [50000, 105000] | Air Pressure |
| $a_c$ | - | [0.1, 1.0] | Accomodation Coefficient |

**Table 1. Pyrcel parcel model input parameters and sampling range used for emulator training.**

## 2.1 Machine Learning Techniques

We assess three commonly used machine learning regression models: ridge regression, gradient boosted regression trees, and deep neural networks. All models take the quantities in Table 1 as inputs and predict the activated fraction of aerosol, which ranges from 0 to 1.

Ridge regression is a linear prediction technique that optimizes coefficients using a penalized cost function that aims to account for and reduce the impact of collinearity in the training dataset. This is done through the addition of an L2 penalty to the commonly used sum of square residual minimization from ordinary least squares fitting. Ridge regression has been used in a variety of prediction tasks in the Earth System Sciences including ozone chemistry (Nowack et al., 2018 ) and estimating the climate sensitivity (Bretherton and Caldwell, 2020). In this study, we use the implementation of ridge regression in the glmnet package in the R language (Friedman et al., 2010). For more information on ridge regression, see Hastie et al. (2001).

Gradient boosted regression trees are a class of machine learning algorithm that trains an ensemble of small tree-based regression models. After the first ensemble member is trained, each following member is fit to the residuals of the previous model, and this process is iteratively completed until satisfactory model performance is achieved (e.g. no additional improvement in prediction skill on the validation dataset is gained by adding ensemble members). We specifically use the XGBoost library, as implemented in the XGBoost package in the R language (Chen and Guestrin, 2016). XGBoost has shown to effectively train useful models in the earth sciences, including applications to atmospheric composition (Ivatt and Evans, 2020; Silva et al., 2020b) and evapotranspiration (Fan et al., 2018). For more information on boosted trees and XGBoost see Chen & Guestrin (2016).

Deep neural networks (DNNs) are the third class of machine learning algorithm we explore in this work. DNNs are a type of artificial neural network, with multiple layers between the input and output nodes. In recent years, DNNs have seen widespread
use in the Earth System Sciences as they perform quite well in estimation tasks and scale well on large supercomputing systems, making them ideal candidates for process emulation in models of the Earth System (e.g. Rasp et al., 2018; Silva et al., 2019). We use the Keras library and the TensorFlow implementation in the Python programming language to design and train the DNNs used in this work (Chollet and others, 2015; Martín Abadi et al., 2015). All DNNs here are feed forward neural networks, with each densely connected layer followed by a dropout layer. For more information on DNNs, see Goodfellow et al. (2016).

**2.2 Physical Regularization**

We investigate the improvement to emulator performance achieved by the application of physical regularization. In the context of this work, physical regularization is the process of adding physical information into an otherwise physically naïve machine learning model to help reduce overfitting. The governing hypothesis here is that by including additional physical information, the model should perform better on an unknown test dataset. To that end, we use the maximum supersaturation and activation
fraction parameterizations described in Twomey (1959) (hereafter, Twomey) and Abdul-Razzak & Ghan (2000) (hereafter, ARG) as regularizing terms for all three machine learning methods described here.

The Twomey scheme was developed as a simple expression of only updraft velocity, where the maximum supersaturation in an air parcel is defined as:

$$S_{max} = \left( \frac{1.63 \times 10^{-3} V^{\frac{3}{2}}}{ck\beta\left(\frac{3}{2}, \frac{k}{2}\right)} \right)^{\frac{1}{k+2}} \tag{1}$$

And the activated fraction is:

$$ActFrac_{Twomey} = \frac{cS_{max}^{k}}{N} \tag{2}$$

Where V is the vertical velocity (cm/s), c and k are fitted parameters (here c=2000 and k=0.4), $\beta$ is the beta function (evaluated here: 4.48), and N is the aerosol number concentration in the air parcel (see Twomey, 1959 for more details). We bound Equation 2 within the range of 0 and 1, to account for known limitations in the scheme (e.g. Ghan et al., 2011). We use Equation
1 and Equation 2 as regularizing terms through a simple hybrid modelling approach where the machine learning emulator is optimized to calculate the residual of the parcel model from the original Twomey (1959) estimates. This is visualized in the flowchart diagram in Figure 1. Stated mathematically, we calculate:

$$ActFrac = ActFrac_{Twomey} + f_{ActFrac}(x) \tag{3}$$

Where $ActFrac$ is the target parcel model activation fraction to emulate, $ActFrac_{Twomey}$ is the estimate from the Twomey scheme, $f_{ActFrac}(x)$ is the function the machine learning emulators will be trained to learn, and $x$ is the set of input parameters. This method allows for some of the nonlinear behavior of $ActFrac$ to be encoded into the estimation prior to any machine learning optimization, and thus should potentially allow for a better solution to this ill-posed estimation task. We additionally feed the emulators with the Twomey predicted $S_{max}$ and $ActFrac_{Twomey}$ as additional input variables for the prediction tasks.

The calculation of the maximum supersaturation and activated fraction in the ARG scheme is more involved than the Twomey scheme and described in detail in Abdul-Razzak & Ghan (2000). We incorporate the ARG estimated activation fraction identically to the Twomey regularization, through learning the residual of the ARG scheme and the parcel model. As with the Twomey regularization, we feed the ARG regularized emulators the ARG predicted $S_{max}$ and activated fraction as additional input variables for the prediction task. The impact of including the ARG or Twomey parameterization predicted $S_{max}$ and activated fraction is marginal in terms of net performance of the emulator, though since the information is already calculated in the regularization step, including it in the model input space adds extra information for little computational cost.

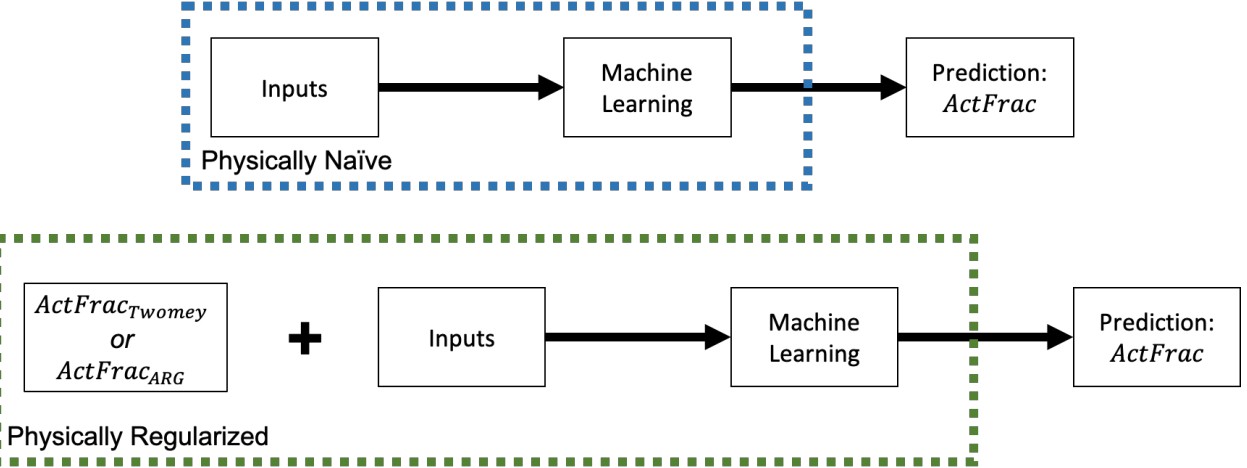

**Figure 1. Schematic diagram of the emulator construction for the physically naïve and physically regularized emulators.**

## 3 Emulator Training

We generated a training and evaluation dataset of 20,000 realizations of the detailed Pyrcel parcel model using the range of environmental conditions summarized in Table 1, sampled using a Latin Hypercube Sampling (LHS) technique as implemented in the SMT python package (Bouhlel et al., 2019). 10,000 simulations were completed with LHS given the limits

in Table 1. 10,000 additional samples were completed using the same LHS limits in Table 1 without the logarithmic transformation applied to the vertical velocity, aerosol number density, or aerosol mean diameter. This sampling method and input parameter space is similar to previous aerosol activation parameterization development datasets using the Pyrcel parcel model (Rothenberg and Wang, 2015). The Pyrcel model fails to converge in the numerical solver for one case out of the 20,000 total simulations during certain conditions unlikely to occur in the real atmosphere (very low pressure with high temperatures and updraft velocities), this case was removed from the training dataset. The full dataset was randomly split into training (70%), validation (10%), and testing (20%) datasets. The training dataset was used to optimize the machine learning model parameters and hyperparameters as assessed by evaluation against the validation datasets. Final model performance was assessed based on performance on the test dataset. We completed an additional set of 1,000 simulations using the same input space in Table 1 and Latin Hypercube Sampling, but ranging from 310 to 314 K, 4K warmer than the training dataset. This is intended to assess model generalizability, or performance on out of sample training data. All features were standardized through a Z-score normalization where the mean was subtracted from each feature, followed by dividing each feature by its standard deviation.

## 3.1 Hyperparameter Selection

All three of the emulator methods used in this work require some degree of hyperparameter selection within the model architecture. Unless otherwise stated, we used package default values for all hyperparameters. Hyperparameters were selected separately for each application of the emulators in this work based on validation dataset performance and are summarized in Table 2. In general, we found that the emulator performance was not particularly sensitive to the hyperparameter tuning; the performance metrics only improved marginally after more optimal parameters were selected (not shown).

In ridge regression, the strength of the L2 norm penalty is controlled by a hyperparameter, commonly written "lambda". We selected this lambda exponential value through directly searching across a range of 101 values from -2 to 3 in increments of 0.05. For the cases investigated here, the validation error tended to asymptotically decrease with smaller lambda values below approximately -1.1.

The XGBoost hyperparameters chosen here were the learning rate, the maximum depth of each tree, and the total number of trees included in the emulator. We searched across these parameters through a grid search of the learning rate and the maximum depth, spanning 0.1 to 0.9 in steps of 0.1 for the learning rate, and 2 to 24 in steps of 2 for the maximum tree depth. For all trees, we allowed the trees to continuously grow with 25 early stopping rounds determining the final depth. Once adding trees does not improve the performance of the emulator on the validation dataset for 25 tree additions, the model training is stopped.

The DNN hyperparameter tuning requires a more careful approach than direct grid search due to the very large hyperparameter optimization space. Our approach involves using automated hyperparameter tuning software to suggest candidate model

hyperparameters, and then fully evaluating the top candidates individually. For automated hyperparameter tuning, we used the Keras Tuner software package, which searches over a wide range of possible hyperparameters more efficiently than random or grid search techniques (O'Malley et al., 2019). Keras Tuner parameters included the hyperband search algorithm, a validation loss objective, with maximum tuner and training epochs of 100, and a factor value of 3. We chose to search across the number of layers, the number of nodes per layer, the dropout rate following each dense layer, and the DNN learning rate. We allowed up to 5 network hidden layers, each with between 10 and 100 nodes, a dropout rate of 0.1 to 0.9, and a learning rate of $10^{-2}$, $10^{-3}$, or $10^{-4}$. We used the Adam optimizer, and ReLu activation functions for all but the output layer, which was set to linear function for all prediction tasks. We then took the top 15 suggested model configurations from the Keras Tuner search, fully trained them, and selected the best performer from that subset. For these fully trained models, we use a batch size of 64 and optimize for the number of training epochs using early stopping, with 25 early stopping epochs.

| | Naïve | Twomey-Regularized | ARG-Regularized |
|---|---|---|---|
| **Ridge** | | | |
| Lambda | -1.9 | -2.0 | -2.0 |
| | | | |
| **XGBoost** | | | |
| Max Depth | 8 | 8 | 6 |
| Eta | 0.1 | 0.1 | 0.1 |
| | | | |
| **DNN** | | | |
| Learning Rate | 1.00E-03 | 1.00E-04 | 1.00E-03 |
| Training Epochs | 40 | 147 | 125 |
| # of Layers | 3 | 3 | 3 |
| # of Nodes | [100, 80, 40] | [100, 40, 70] | [50, 100, 30] |
| Dropout Fraction | [0.3, 0.1, 0.5] | [0.1, 0.1, 0.2] | [0.1, 0.3, 0.2] |

**Table 2. Emulator hyperparameters chosen in this study.**

## 4 Emulator Evaluation

We evaluate the skill of these emulators in reproducing the activation fraction prediction within the test set, as described in Section 3. As machine learning predictive skill on the training set is not always an indicator of predictive skill on the test set, we discuss only test set performance here as a more strict evaluation criteria.

### 4.1 Physically Naïve Machine Learning Emulators

The test set performance of the activation fraction machine learning emulators without any physical regularization, here
referred to as "physically naïve", is summarized in Figure 2 along with the ARG parameterization predictions. In general, the
emulators perform well, with the majority of points clustered around the 1:1 line, mean squared errors (MSE) below 0.05, and
high $R^2$ values (above ~0.7). The best performance comes from the DNN and the XGBoost regressions, followed by the ridge
regression. We additionally include the ARG activation parameterization in Figure 2 as a baseline performance comparison
with a commonly used existing activation parameterization. Both the DNN and the XGBoost regression outperform the ARG
benchmark parameterization, with lower mean squared errors and higher $R^2$ values. The Twomey scheme is not shown in
Figure 2, as it performs relatively quite poorly (MSE = 0.29, $R^2$=0.03) and is thus not a particularly useful benchmark as
compared to the relatively skillful ARG parameterization.

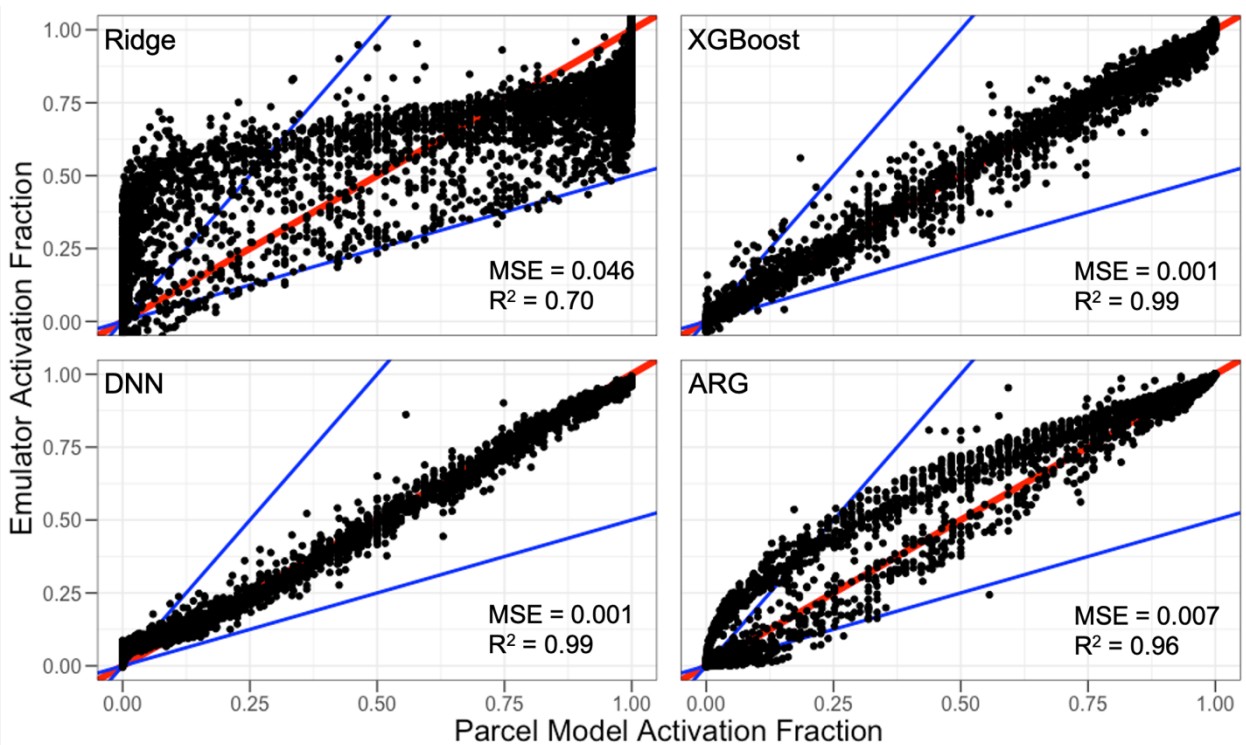

**Figure 2. Scatterplot comparisons of the three physically naïve machine learning emulators and the ARG scheme predicted**
**activation fraction with the detailed parcel model. The 1:1 line is in red, and the blue lines represent a factor of two difference.**
**Performance statistics are on each figure panel.**

For certain cases very near the mass-conserving bounds of 0 to 1 (~10% of the test data), the emulators  predict activation
fraction values that extend beyond those bounds. Other than for the linear ridge regression, these deviations outside of the
mass-conserving bounds are all very small (less than 0.01). Though imposing that range as an upper and lower bound on those

regressions would be a sensible choice if the emulators were implemented into an Earth System Model, the imposition does not substantially impact performance metrics (MSE and $R^2$).

For the DNN emulator, we chose a linear activation function for the final model layer. Since the activation fraction varies from 0 to 1, a sigmoid activation function would also be a logical choice and would encode a small amount of prior information into the system. However, in this case, the linear activation function has slightly better predictive skill. Using a sigmoid activation

function does not appreciably change the results shown here, and actually leads to slightly worse emulator performance.

Machine learning emulators tend to improve performance with larger training datasets. In this case, training using only half of the available data still leads to relatively skillful emulators. For example, the same DNN trained on 50% of the training samples has an MSE of 0.0017 and an $R^2$ of 0.99. This is worse than the DNN in Figure 2, which is fully trained, but does still outperform the commonly used and physically based ARG parameterization.

**4.2 Physically Regularized Emulators**

Including physical regularization generally improves model performance on the test set. Performance for the Twomey and ARG regularized models is shown in the scatterplots in Figure 3. Ultimately, the poor performance of the Twomey scheme prior to implementation as a regularizing term limits the added value it provides to the emulators. The performance gains by Twomey regularization as compared to the physically naïve emulators are generally ~10% or less in terms of mean squared

error for the emulators, with differences in $R^2$ values of generally less than a few percent. While this specific performance gain is not large, the Twomey scheme can be calculated as a simple power of vertical velocity and is thus a computationally simple technique for improving emulator accuracy.

The benefits of the ARG regularization are larger and more consistent across emulators, as can be seen in Figure 3. For all three machine learning model types, the ARG regularization performs the best, with the lowest mean squared error and highest

$R^2$ values within each emulator category. While all model types improve with the additional information provided by the ARG scheme, the smallest relative improvement is that of the DNN and XGBoost emulators, and the largest is for the ridge regression. The linear ridge regression, when regularized by the ARG scheme, outperforms the standard ARG parameterization with a 40% relative reduction in the mean squared error. Framing this finding from the perspective of the ARG scheme, a linear correction term with ridge-calculated coefficients could reduce the parameterization error by 40%, and nonlinear

corrections (i.e., XGBoost or DNNs) could further reduce that error by an order of magnitude. As with the naïve emulators, for predictions very near the bounds of 0 to 1, the physically regularized emulators do tend to predict variables outside of that range. The linear ridge regression predicts the largest deviations, where the physically regularized XGBoost and DNN models typically predict deviations within 0.01 of the bounds.

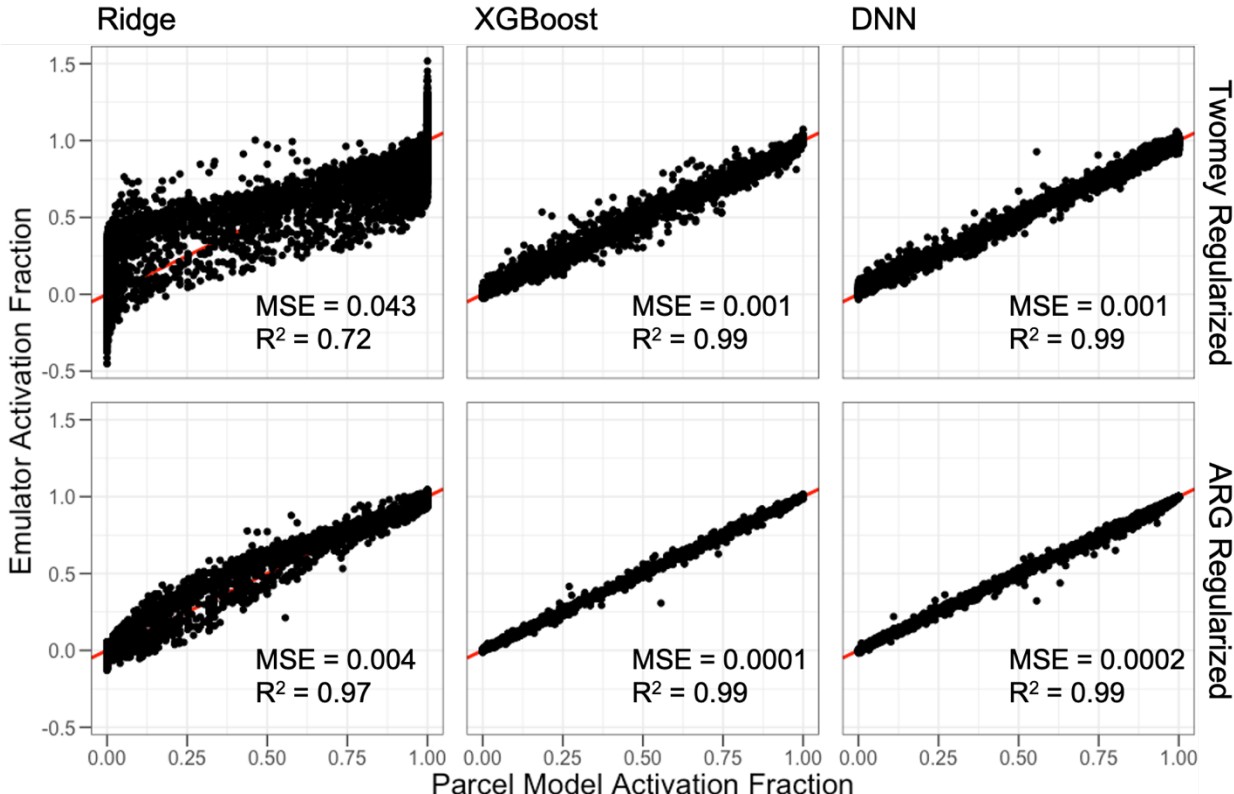

**Figure 3. Scatter plots for the various emulator types against the parcel model activation fraction. The 1:1 line is shown in red, and emulator specific performance statistics are shown in each panel.**

Physical regularization particularly improves the emulator behavior for very low activation fractions. As an example, Figure 4 shows all three versions of the DNN emulator performance for cases with activation fractions below 0.1. The physically naïve emulator substantially overestimates most very low parcel model simulated activation fractions. The regularization from the Twomey scheme improves upon this issue but increases the emulator scatter in this range. The more detailed and general ARG regularization reduces the over prediction issue from the naïve scheme even further, with the best overall performance. This potentially has implications for the impact of these emulators when implemented into an Earth System Model, where low activation fractions can be common.

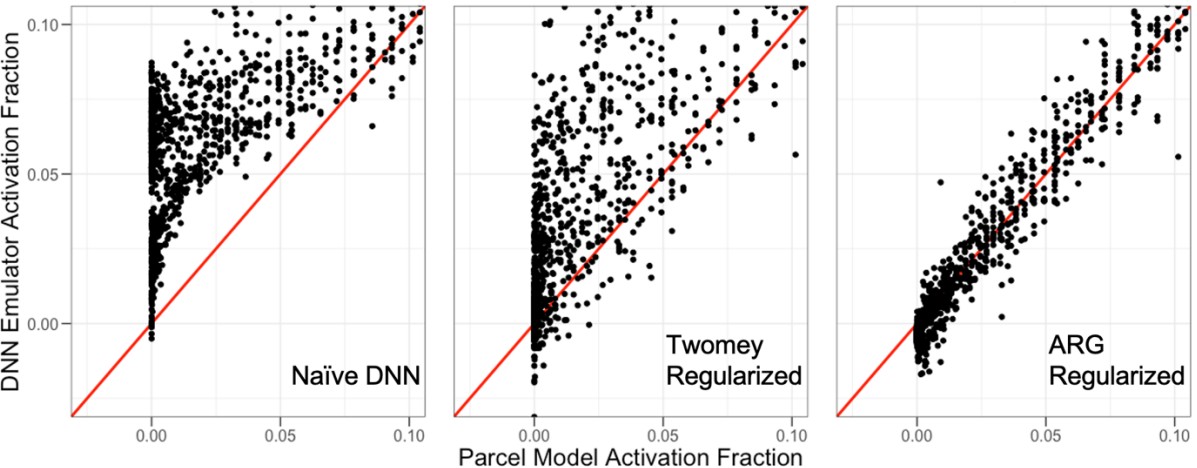

**Figure 4. Scatter plots for the various DNN emulator types against the parcel model activation fraction for cases with activation fractions below 0.1. The 1:1 line is shown in red.**

The capacity of these emulators (their ability to emulate arbitrarily complex functions) increases with the emulator complexity and number of parameters. As the capacity increases, the benefit of the Twomey and ARG physical regularization decreases. This is evident by the large gains in accuracy when the ridge regression, which is fundamentally a linear model, is physically

regularized as compared to the very modest absolute accuracy gains in the largely non-linear DNNs. To further illustrate this point, we ran sensitivity experiments with the XGBoost emulators, evaluating the prediction error on the validation dataset as a function of the number of boosting iterations for a naïve and an ARG physically regularized emulators, both with the same hyperparameters. Each boosting iteration has the potential to add trees to the model, and thus increases to the emulator capacity. Results are shown in Figure 5. As expected, additional trees (boosting iterations) reduce the mean squared prediction error of

both the naïve and ARG regularized emulators. When the emulator capacity is relatively low (the number of trees is low), the physically regularized emulator is much more skillful in terms of MSE. As capacity increases this regularization accuracy benefit is reduced substantially, though it is always present to some extent. For a given machine learning technique, increased capacity typically comes with increased computational cost. Including physical information through physical regularization can thus be a computationally efficient strategy for achieving a given model accuracy with lower capacity.

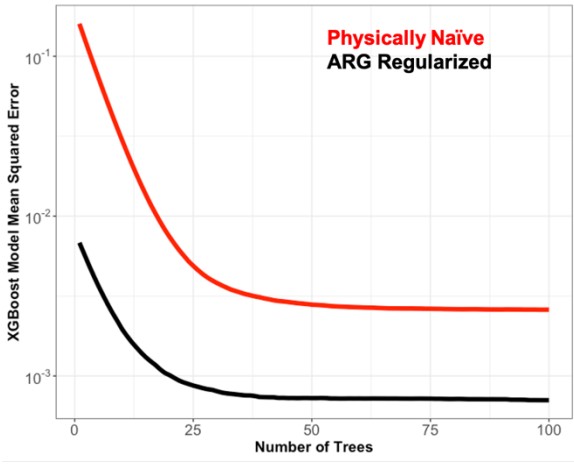


**Figure 5. Mean Squared Error as a function of the XGBoost number of trees for both the naïve non-regularized emulator (red) and the ARG regularized emulator (black).**

### 4.3 Emulator Generalizability

We further evaluate the emulators using a data set with up to 4K warmer temperatures than used in the training data. This
evaluation on input data outside of the training parameter space can provide useful information on the generalizability of the emulators, and their performance when used in scenarios that may be not well characterized by the training dataset (potentially likely in a climate model simulation). The generalizability test dataset was generated using the same parameter space as the training dataset described in Table 1, except the temperature range was from 310K to 314K.

The emulators tend to perform fairly well on this generalizability test, with MSE and $R^2$ values similar to the performance
shown on the test dataset. Summary results are shown for the DNN emulator for all three emulator designs in Figure 6. The results in Figure 6 are qualitatively consistent for the ridge and XGBoost emulators. The best emulator performance is from the ARG regularized emulator, followed by the Twomey regularization, and then the physically naïve DNN. Ultimately, the limited conditions under which the original Twomey (1959) formulation is derived (e.g. Ghan et al., 2011) limit the generalizability performance of the scheme as a regularizing term for these emulators. It is important to note that though the
emulators perform well in this generalizability test, there is no guarantee that they will perform well for all extrapolation cases, particularly those that deviate very far from the training data parameter space.

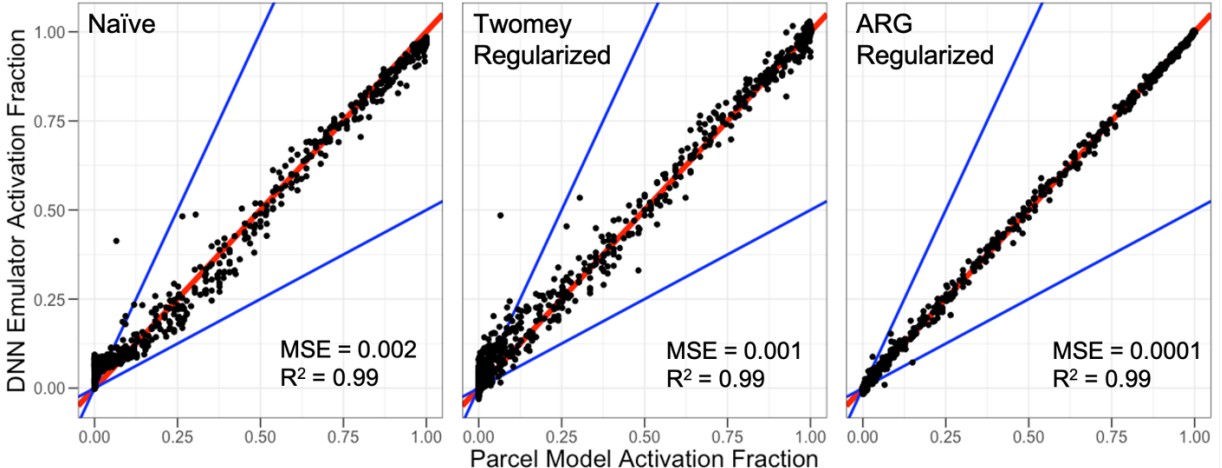

**Figure 6. Activation fraction performance of the DNN emulators used here on the +4K generalizability test dataset. The 1:1 line is in red and the blue lines represent a factor of two difference. Performance statistics are on each figure panel.**

### 4.4 Emulator Sensitivity

We additionally evaluate the emulators as a function of variability in their input parameter space. Analogous to Rothenberg and Wang (2015), we fix all but one input parameter and explore the variability in the emulator as a function of one single input parameter. Results for the DNN emulators as a function of number concentration, vertical velocity, mean radius, and hygroscopicity are shown in Figure 7. Other emulators are generally consistent, with worse overall skill for the ridge regression emulators. Generally, the emulators all perform well. The best performance is associated with the ARG regularized scheme, and the most aberrant performance from the Twomey regularization.

The emulators are all within ~10% for all predictions as a function of number concentration, mean radius, and vertical velocity. Much larger errors are apparent for emulator performance in cases with very low aerosol hygroscopicity, where the only skillful emulator is the ARG regularized model. In the real atmosphere, very low hygroscopicity values are reasonably common, and activation overestimates by nearly a factor of 4 for the naïve and Twomey regularized schemes would likely have a substantial impact on climate, producing too many cloud droplets by activating hydrophobic aerosols. These issues are consistent with the large overestimates seen at low activations in Figure 4. Though the specific issue of the poor performance of the Twomey regularized and naïve emulators in this low hygroscopicity range could potentially be somewhat resolved with additional model training data and other training optimization techniques (e.g., transfer learning on a subsample of the data, optimizing in log space, etc.), initial tests suggest that none of these issues completely solve the performance issues. This strongly motivates the use of sufficient physical regularization to address other potentially unknown biases in emulator performance.

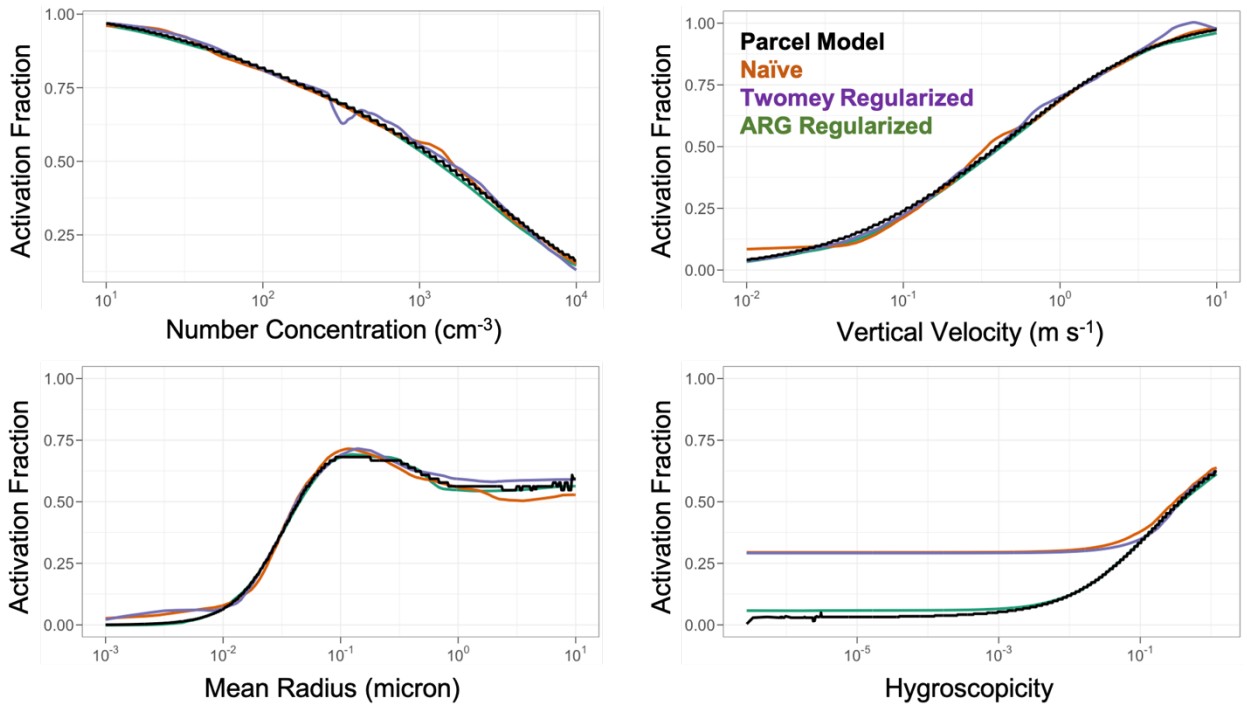

**Figure 7. Variability in predicted activation fraction from the DNN emulators and the parcel model as a function of input parameters. Number concentration, vertical velocity, mean radius, and aerosol hygroscopicity are shown here. The parcel model is shown in black, the naïve DNN emulator in orange, the Twomey regularized emulator in purple, and the ARG regularized emulator in green. For each panel, all other parameters are fixed at: number concentration = 1000 cm$^{-3}$, mean radius = 0.05, aerosol mode standard deviation = 1.8, hygroscopicity = 0.54, vertical velocity = 0.5 m s$^{-1}$, Temperature = 283 K, Pressure = 85000 Pa, and an accommodation coefficient of 0.95.**

## 5 Summary and Future Directions

Though aerosol activation can be challenging to simulate with high accuracy and computational speed, machine learning techniques provide a potential path forward. We demonstrate that several classes of machine learning models can produce accurate emulations of a detailed parcel model, competitive with existing model parameterizations. We evaluate the performance of three machine learning regression models: ridge, XGBoost, and DNNs. Both the XGBoost and the DNN regression outperform the commonly used ARG parameterization, with the best overall performance from the DNN.

We show that including physical information in the construction and training of these machine learning models can yield improved emulators through physical regularization with the Twomey (1959) and Abdul-Razzak & Ghan (2000) aerosol activation parameterizations. In particular, improved performance through physical regularization is apparent in emulator edge cases and cases that are poorly represented in the emulator training data. These accuracy gains are dependent on the quality of the physical information provided in the regularization step, and the capacity of the machine learning model. The original

Twomey (1959) activation scheme is limited in scope, and only applicable to certain atmospheric conditions. This leads to reduced performance of the Twomey-regularized emulators over those regularized by the globally applicable ARG parameterization. The improved performance from physical regularization is somewhat dependent on emulator capacity: once sufficient emulator capacity is available, the accuracy differences between physically informed and physically naïve models become small.

Machine learning techniques have been shown to scale quite well on large scale super computing systems, particularly for feedforward deep neural networks like those applied here. This good computational speed scaling lends support to the applicability of machine learning emulators in computationally expensive Earth System Models, like the DOE's Energy Exascale Earth System Model (E3SM). Additionally, the algorithms investigated here (XGBoost and DNNs) have efficient GPU implementations and are thus directly applicable to next generation high performance computing architectures that may

rely more on GPUs. As the representation of processes in Earth System Models grows more complex and computationally expensive, the development and application of novel emulation techniques becomes continually more useful as an important step in model development.

**Acknowledgments**

A portion of the research described in this manuscript was conducted under the Laboratory Directed Research and

Development Program at Pacific Northwest National Laboratory (PNNL), a multiprogram national laboratory operated by Battelle for the U.S. Department of Energy. SJS is grateful for the support of the Linus Pauling Distinguished Postdoctoral Fellowship program. SJS would like to acknowledge his parents for providing childcare during the COVID-19 pandemic which enabled the writing of this manuscript. This study was partly supported by the "Enabling Aerosol-cloud interactions at GLobal convection-permitting scalES (EAGLES)" project (74358), funded by the U.S. Department of Energy, Office of Science,

Office of Biological and Environmental Research, Earth System Model Development (ESMD) program. The Pacific Northwest National Laboratory is operated for the U.S. Department of Energy by Battelle Memorial Institute under contract DE-AC05-76RL01830.

**Author Contributions**

SJS, P-M, and JCH designed the study. SJS developed and implemented the emulator techniques. DR developed the Pyrcel

code. All authors contributed to the paper preparation.

**Code and Data Availability**

The current version of the Pyrcel model is available from the project website: pyrcel.readthedocs.io under the New BSD (3-clause) license. The exact version of the Pyrcel model used to produce the results used in this paper along with the analysis code and model output is archived on Zenodo: http://doi.org/10.5281/zenodo.4319145.

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
