# Peer review of "Physically Regularized Machine Learning Emulators of Aerosol Activation"

_Geoscientific Model Development, 2020_

## Referee Comment (RC1) · Anonymous Referee #1 · 7 Jan 2021

This is an interesting paper reporting an effort of applying three machine learning regression methods, including ridge, gradient boosted regression trees, and all-connected neural networks, to derive a parameterization for aerosol activation that mimics the behavior of a physical parcel model, the Pyrcel model by Rothenberg and Wang (2015), largely in analog to the procedure in Rothenberg and Wang who used a probabilistic collocation method combining with a Latin hypercube sampling. The authors found that when certain parameterization or simplified model of aerosol activation were introduced as an additional constraint in training, the performance of trained machines would be significantly improved. Overall, the authors have demonstrated that, as orchestrated by several other efforts, machine learning algorithms can be powerful tools for deriving or rederiving parameterizations for physical processes. The paper is

well structured. The results are presented in a good order and also informative. Nevertheless, certain points need to be addressed or clarified before the paper can be accepted for publication.

One interesting result shown in Fig.5 is that a high-capacity machine (XGBoost) trained physically naively can simply match the performance of the one trained with physical constraint. From the perspective of statistics, using a physical constraint in training is simply to provide a better defined a poster scope so that the machine can be trained to easily reach a desired performance with low training cost. However, as far as the base model is regarded as the ground truth, a well-performed machine could be trained without such constraint, as demonstrated in Fig.5 credited to the authors. Generally speaking, the performance of a machine learning model could be optimized with increasing capacity, thus a point here worthy a discussion is whether the cost of coupling a simple model or alternative parameterization (likely with a considerable cost) with a low-capacity model would be better than a high-capacity model alone in application. In this sense, a better purpose of using alternative parameterization here seems just evaluating the alternative parameterization itself.

It was shown in the paper that a number of predicted points by the machines exceeded the physical bound of activation ratio of [0, 1] (more evident in Fig.4). In many cases, this type of outcomes might be a result from use of unnormalized multidimensional features. Firstly, the authors might need to mention the number or ratio of these points. Secondly, had the authors tested training with normalized features? If not, what is the specific reason for not doing so?

Rich resources for machine learning nowadays make the task to understand the sensitivity of targeted outcome to input features much easier. Besides the sensitivity study presented, had the authors used functionalities such as feature selection and feature importance to analyze the sensitivity of the performance of trained machines to the features?

[Figure]

Specific comments.

Line 20-25, the sentences could be rearranged to make the arguments lining up more logically, a suggestion is to move "Cloud formation... Seinfeld and Pandis, 2006)" (Ln 20-22) to ahead of "These aerosol-cloud..." (Ln 25) and modified "Cloud formation" to "It is because that cloud formation"; then change Ln 23 "Hobbs, 2006) and by changing" to "Hobbs, 2006). Aerosols can also change".

Line 27: "quite" could be removed.

Line 47, "few observations": did the authors mean "without observations"? If so, the sentence can stand, otherwise, change "few" to "a few".

Line 48, change "few" to "a limited number of".

Line 55, "are unable" to "are still unable".

Line 56, "will longer run times" to "with longer run times"?

Table 1. The caption should include definitions of features, and please change the font and reformat subscript to make them more readable.

Line 118, should use (1) after the equation instead of Equation 1? The same is applied to later equations. Also, please change font size, and also add a space after "," inside beta ().

Line 132, add "with" after "emulator".

Line 257, remove one of the two "in".

Line 286-287, "This strongly...", as discussed in the previous general comment, the key here for training a better performing machine perhaps is to choose an algorithm adequate for the problem, i.e., nonlinear one for a nonlinear problem.

Fig. 7, Results of activation fraction versus hygroscopicity: what would the high-capacity XGBoost model behave?

Line 311-318, the discussion about training with GPU is adequate, however, the type of chip might not be a central issue for applications of trained machines (just a matrix of coefficients) in practice.

———————————————————

---

## Short Comment (SC1) · 7 Jan 2021

We would like to bring into attention the work we have done related to the manuscript topic and which should at least to be cited. In Lipponen et al. (2013), a very similar approach to the presented one was introduced. Lipponen et al. (2013) used an improved hybrid physical-machine learning model to improve the representation of cloud droplet formation in large scale models. A machine learning regression-based emulator model was employed to reduce the approximation error caused both by the employment of ARG-parameterization for cloud droplet activation and low number of aerosol size sections in the SALSA aerosol module. In this manuscript under evaluation, the authors refer to this approach as a physically regularized machine learning emulator. Later, in Lipponen et al. (2018), it was also demonstrated that the approximation error–

corrected simulation improved signiïñĄcantly the accuracy of the reduced model and also outperformed the approach where a statistical learning-based predictor is constructed directly for the accurate model without physical constrains or information from the reduced model.

Lipponen A., V. Kolehmainen, S. Romakkaniemi and H. Kokkola: Correction of approximation errors with random forest applied to modeling of aerosol first indirect effect, Geosci. Model Dev., 6, 2087-2098, doi:10.5194/gmd-6-2087-2013, 2013.

Lipponen, A., J. M. J. Huttunen, S. Romakkaniemi, H. Kokkola, and V. Kolehmainen: Correction of Model Reduction Errors in Simulations, SIAM Journal on Scientific Computing, 40:1, B305-B327, 2018.

---

## Referee Comment (RC2) · Anonymous Referee #2 · 15 Jan 2021

Silva et al. discuss the use of machine learning emulators to predict the activation of aerosol into cloud droplets. For this they test three machine learning algorithms – ridge regression, gradient boosted trees, and deep neural networks – and use them with and without an a priori estimate of aerosol activation based on two physical parameterizations of different complexity (denoted the Twomey scheme and the ARG scheme, respectively). They find that two of the tested machine learning emulators (gradient boosted trees and neural network) outperform the parameterizations even without imposing physical constraints, and all methods outperform the regular parameterization when combined with the physical parameterizations.

The paper is well written and the study is within the scope of the journal. The machine learning approach seems generally sound (see minor points below) and the authors

[Figure]

provide a detailed description of the methodology. I thus recommend the manuscript for publication after addressing the following minor points:

- Table 1: Please provide a long name for each parameter, e.g., temperature, pressure, etc.

- Figure 2: Is there a reason why the performance of the Twomey scheme is not shown? Also, please state in the figure (or at least in the figure label) that these are the results for the physically naïve emulators.

- Section 4: it would be useful to show the machine learning statistics for both the training data and the test data to demonstrate that the models are not suffering from overfitting.

- On line 260, I think it should say: "tend to perform..."

- Section 4.4: The weak performance of the naïve and Twomey regularized emulator under low hygroscopicity regimes seems somewhat surprising. Doesn't this imply that the training data did not include enough training data capturing a low hygroscopicity environment? Based on Table 1, the hygroscopicity range of the training data spans 0 to 1.2, so the training data should capture this range at least to some extent. Or does this result suggest that the hygroscopicity value should be log-transformed to give higher weights to the lower bound? Given that low hygroscopicity values are not uncommon in the real atmosphere, this should be addressed a bit more convincingly in the revised version of the manuscript.

- Figure 7: is it possible to also show the performance of the regular parameterizations (without any machine learning)? This would help demonstrate the value of adding the machine learning correction to these parameterizations.

[Figure]

---

## Author Comment (AC1) · 26 Mar 2021

We thank the reviewers for their thoughtful comments on the manuscript and have addressed them in detail below. Reviewer comments are in *blue italics*, and our responses are in black normal text.

**Response to Reviewer #1**

*One interesting result shown in Fig.5 is that a high-capacity machine (XGBoost) trained physically naively can simply match the performance of the one trained with physical constraint. From the perspective of statistics, using a physical constraint in training is simply to provide a better defined a poster scope so that the machine can be trained to easily reach a desired performance with low training cost. However, as far as the base model is regarded as the ground truth, a well-performed machine could be trained without such constraint, as demonstrated in Fig.5 credited to the authors. Generally speaking, the performance of a machine learning model could be optimized with in- creasing capacity, thus a point here worthy a discussion is whether the cost of coupling a simple model or alternative parameterization (likely with a considerable cost) with a low-capacity model would be better than a high-capacity model alone in application. In this sense, a better purpose of using alternative parameterization here seems just evaluating the alternative parameterization itself.*

The reviewer makes a good point that there is an inherent tradeoff between the capacity and skill of these methods, which does have implications for computational cost. We have updated the manuscript to include this in the discussion of Figure 5 on lines 283-285:

"For a given machine learning technique, increased capacity typically comes with increased computational cost. Including physical information through physical regularization can thus be a computationally efficient strategy for achieving a given model accuracy with lower capacity."

The reviewer has pointed out an issue in our communication of the results summarized Figure 5. While the additional model capacity does certainly improve the performance of these machine learning techniques, the physically regularized model always has better skill. We have updated the figure to now be in log-scale to better illustrate this result.
The difference between the two figures is shown below:

[Figure]

[Figure]

*It was shown in the paper that a number of predicted points by the machines exceeded the physical bound of activation ratio of [0, 1] (more evident in Fig.4). In many cases, this type of outcomes might be a result from use of unnormalized multidimensional features. Firstly, the authors might need to mention the number or ratio of these points. Secondly, had the authors tested training with normalized features? If not, what is the specific reason for not doing so?*

We agree with the reviewer that normalization can be an important step in the machine learning model development. We do normalize the features in this work, and update the text to be explicit in that sense on line 174:

"All features were standardized through a Z-score normalization where the mean was subtracted from each feature, followed by dividing each feature by its standard deviation."

The ratio of points outside of [0,1] is now stated in the text as well, on lines 224-226:

"For cases very near the mass-conserving bounds of 0 to 1 (~10% of the test data), the emulators all predict activation fraction values that extend beyond those bounds. Other than for the linear ridge regression, these deviations outside of the mass-conserving bounds are all very small (less than 0.01)."

While these unphysical predictions are certainly an important caveat regarding model skill, the bounds of [0,1] are reasonably easy to enforce in a large-scale modeling framework (e.g. through clipping).

*Rich resources for machine learning nowadays make the task to understand the sen- sitivity of targeted outcome to input features much easier. Besides the sensitivity study presented, had the authors used functionalities such as feature selection and feature importance to analyze the sensitivity of the performance of trained machines to the features?*

The reviewer raises an interesting point about the potential value of interpretability methods in the development of machine learning emulators. We did not apply them in detail in this work for several reasons. Primarily, we begin this analysis with specific and concrete knowledge of which features contribute to the prediction (i.e., the parent data generating model is known). Additionally, we have a relatively small number of features to begin with, so feature selection techniques for subset selection and computational efficiency were not a design requirement. Lastly, the majority of these selection and importance algorithms (e.g., Layer-wise Relevance Propagation, XGBoost Importance/Gain, etc.) are specific to a given machine learning technique and cannot be easily compared across the DNN, XGBoost, and Ridge methods used in this work.

We show an example of the XGBoost importance metrics below, specifically the gain metric. This demonstrates that while all features are important for prediction (values > 0), certain features that the parent model is most sensitive to (e.g. vertical velocity and aerosol population parameters) are more prominent in the XGBoost model as well.

[Figure]

**Specific comments**

*Line 20-25, the sentences could be rearranged to make the arguments lining up more logically, a suggestion is to move "Cloud formation. . . Seinfeld and Pandis, 2006)" (Ln 20-22) to ahead of "These aerosol-cloud. . ." (Ln 25) and modified "Cloud formation" to "It is because that cloud formation"; then change Ln 23 "Hobbs, 2006) and by changing" to "Hobbs, 2006). Aerosols can also change".*
Done. Thank you.

*Line 27: "quite" could be removed.*
Removed.

*Line 47, "few observations": did the authors mean "without observations"? If so, the sentence can stand, otherwise, change "few" to "a few".*
Updated to "a few".

*Line 48, change "few" to "a limited number of".*
Done.

*Line 55, "are unable" to "are still unable".*
Corrected.

*Line 56, "will longer run times" to "with longer run times"?*
Updated.

*Table 1. The caption should include definitions of features, and please change the font and reformat subscript to make them more readable.*
Updated.

*Line 118, should use (1) after the equation instead of Equation 1? The same is applied to later equations. Also, please change font size, and also add a space after "," inside beta ().*
Updated, thank you.

*Line 132, add "with" after "emulator".*
Added.

*Line 257, remove one of the two "in".*
Done.

*Line 286-287, "This strongly. . .", as discussed in the previous general comment, the key here for training a better performing machine perhaps is to choose an algorithm adequate for the problem, i.e., nonlinear one for a nonlinear problem.*
Agreed.

*Fig. 7, Results of activation fraction versus hygroscopicity: what would the high- capacity XGBoost model behave?*
The high-capacity XGBoost model behaves similarly to the physically regularized XGBoost model, with larger errors.

*Line 311-318, the discussion about training with GPU is adequate, however, the type of chip might not be a central issue for applications of trained machines (just a matrix of coefficients) in practice.*
We agree that the training gains are larger on GPUs than the application of already trained models. We keep the statement general to account for other complexities in the potential machine learning pipeline (e.g. online learning).

---

## Author Comment (AC2) · 26 Mar 2021

We thank the reviewers for their thoughtful comments on the manuscript and have addressed them in detail below. Reviewer comments are in *blue italics*, and our responses are in black normal text.

**Response to Reviewer #2**

*Table 1: Please provide a long name for each parameter, e.g., temperature, pressure, etc.*
Updated.

*Figure 2: Is there a reason why the performance of the Twomey scheme is not shown? Also, please state in the figure (or at least in the figure label) that these are the results for the physically naïve emulators.*
The main purpose of Figure 2 was to show the performance of the naïve emulators, with the ARG scheme only shown as a reference. The Twomey scheme skill is very bad, with orders of magnitude higher error and worse $R^2$, as stated on Line 215. As such, we chose not to include the Twomey performance, as it is would distract from the main purpose of the figure. We have updated lines 218-219 to more specifically address this:
        "The Twomey scheme is not shown in Figure 2, as it performs relatively quite poorly (MSE = 0.29, $R^2$=0.03) and is not a particularly useful benchmark as compared to the relatively skillful ARG parameterization."

*Section 4: it would be useful to show the machine learning statistics for both the training data and the test data to demonstrate that the models are not suffering from overfitting.*
The reviewer raises a good point, that predictive skill on the training set is not necessarily indicative of skill on the test set predictions. We are more explicit in the manuscript on lines 207-210 that all evaluation is done on the test set to more accurately represent the lack of strong overfitting in our models:
        "We evaluate the skill of these emulators in reproducing the activation fraction prediction within the test set, as described in Section 3. As machine learning predictive skill on the training set is not always an indicator of predictive skill on the test set, we discuss only test set performance here as a more strict evaluation criteria."

*On line 260, I think it should say: "tend to perform. . ."*
Updated, thank you.

*Section 4.4: The weak performance of the naïve and Twomey regularized emulator under low hygroscopicity regimes seems somewhat surprising. Doesn't this imply that the training data did not include enough training data capturing a low hygroscopicity environment? Based on Table 1, the hygroscopicity range of the training data spans 0 to 1.2, so the training data should capture this range at least to some extent. Or does this result suggest that the hygroscopicity value should be log-transformed to give higher weights to the lower bound? Given that low hygroscopicity values are not uncommon in the real atmosphere, this should be addressed a bit more convincingly in the revised version of the manuscript.*
The weak performance of the naïve and Twomey regularized emulators indeed might be due to poor representation in the training data. We discuss this further on lines 321-325, however in

the original text it was not apparent that this discussion was specifically toward low hygroscopicity values. We have updated the text:

"Though the specific issue of the poor performance of the Twomey regularized and naïve emulators in this low hygroscopicity range could potentially be somewhat resolved with additional model training data and other training optimization techniques (e.g., transfer learning on a subsample of the data, optimizing in log space, etc.), initial tests suggest that none of these issues completely solve the performance issues."

*Figure 7: is it possible to also show the performance of the regular parameterizations (without any machine learning)? This would help demonstrate the value of adding the machine learning correction to these parameterizations.*

To maintain an easy-to-read figure, we have included only the machine learning-based parameterizations, as the sensitivities of the Twomey and ARG schemes have been described in detail in previous work. We have included a figure with the original parameterizations below.

[Figure]

---

## Author Comment (AC3) · 26 Mar 2021

Dear Sami Romakkaniemi,

Thank you for directing our attention to this prior work. We agree that it should be cited in our manuscript, and have updated the work accordingly on lines 113-114.

Thank you,

Sam Silva

---

## Author Response (AR2)

Dear Editor,

Thank you for the comments and minor revisions. We apologize for accidentally removing the code and data availability section, it was an oversight in editing and responding to the reviewers.

We have updated the code and data availability section to the following statement:

The current version of the Pyrcel model is available from the project website: pyrcel.readthedocs.io under the New BSD (3-clause) license. The exact version of the Pyrcel model used to produce the results used in this paper along with the analysis code and model output is archived on Zenodo: http://doi.org/10.5281/zenodo.4319145.

Thank you,
Sam Silva